# Unconstrained Monotonic Neural Networks

**Antoine Wehenkel**
University of Liège

**Gilles Louppe**
University of Liège

## Abstract

Monotonic neural networks have recently been proposed as a way to define invertible transformations. These transformations can be combined into powerful autoregressive flows that have been shown to be universal approximators of continuous probability distributions. Architectures that ensure monotonicity typically enforce constraints on weights and activation functions, which enables invertibility but leads to a cap on the expressiveness of the resulting transformations. In this work, we propose the Unconstrained Monotonic Neural Network (UMNN) architecture based on the insight that a function is monotonic as long as its derivative is strictly positive. In particular, this latter condition can be enforced with a free-form neural network whose only constraint is the positiveness of its output. We evaluate our new invertible building block within a new autoregressive flow (UMNN-MAF) and demonstrate its effectiveness on density estimation experiments. We also illustrate the ability of UMNNs to improve variational inference.

## 1 Introduction

Monotonic neural networks have been known as powerful tools to build monotone models of a response variable with respect to individual explanatory variables [Archer and Wang, 1993, Sill, 1998, Daniels and Velikova, 2010, Gupta et al., 2016, You et al., 2017]. Recently, strictly monotonic neural networks have also been proposed as a way to define invertible transformations. These transformations can be combined into effective autoregressive flows that can be shown to be universal approximators of continuous probability distributions. Examples include Neural Autoregressive Flows [NAF, Huang et al., 2018] and Block Neural Autoregressive Flows [B-NAF, De Cao et al., 2019]. Architectures that ensure monotonicity typically enforce constraints on weight and activation functions, which enables invertibility but leads to a cap on the expressiveness of the resulting transformations. For neural autoregressive flows, this does not impede universal approximation but typically requires either complex conditioners or a composition of multiple flows.

Nevertheless, autoregressive flows defined as stacks of reversible transformations have proven to be quite efficient for density estimation of empirical distributions [Papamakarios et al., 2019, 2017, Huang et al., 2018], as well as to improve posterior modeling in Variational Auto-Encoders (VAE) [Germain et al., 2015, Kingma et al., 2016, Huang et al., 2018]. Practical successes of these models include speech synthesis [van den Oord et al., 2016, Oord et al., 2018], likelihood-free inference [Papamakarios et al., 2019], probabilistic programming [Tran et al., 2017] and image generation [Kingma and Dhariwal, 2018]. While stacking multiple reversible transformations improves the capacity of the full transformation to represent complex probability distributions, it remains unclear which class of reversible transformations should be used.

In this work, we propose a class of reversible transformations based on a new Unconstrained Monotonic Neural Network (UMNN) architecture. We base our contribution on the insight that a function is monotonic as long as its derivative is strictly positive. This latter condition can be enforced with a free-form neural network whose only constraint is for its output to remain strictly positive.

We summarize our contributions as follows:

- We introduce the Unconstrained Monotonic Neural Network (UMNN) architecture, a new reversible scalar transformation defined via a free-form neural network.

- We combine UMNN transformations into an autoregressive flow (UMNN-MAF) and we demonstrate competitive or state-of-the-art results on benchmarks for normalizing flows.

- We empirically illustrate the scalability of our approach by applying UMNN on high dimensional density estimation problems.

## 2 Unconstrained monotonic neural networks

Our primary contribution consists in a neural network architecture that enables learning arbitrary monotonic functions. More specifically, we want to learn a strictly monotonic scalar function $F(x; \boldsymbol{\psi}) : \mathbb{R} \to \mathbb{R}$ without imposing strong constraints on the expressiveness of the hypothesis class. In UMNNs, we achieve this by only imposing the derivative $f(x; \boldsymbol{\psi}) = \frac{\partial F(x; \boldsymbol{\psi})}{\partial x}$ to remain of constant sign or, without loss of generality, to be strictly positive. As a result, we can parameterize the bijective mapping $F(x; \boldsymbol{\psi})$ via its strictly positive derivative $f(x; \boldsymbol{\psi})$ as

$$F(x; \boldsymbol{\psi}) = \int_0^x f(t; \boldsymbol{\psi}) \, \mathrm{d}t + \underbrace{F(0; \boldsymbol{\psi})}_{\beta}, \tag{1}$$

where $f(t; \boldsymbol{\psi}) : \mathbb{R} \to \mathbb{R}_+$ is a strictly positive parametric function and $\beta \in \mathbb{R}$ is a scalar. We make $f$ arbitrarily complex using an unconstrained neural network whose output is forced to be strictly positive through an ELU activation unit increased by 1. $\boldsymbol{\psi}$ denotes the parameters of this neural network.

**Forward integration** The forward evaluation of $F(x; \boldsymbol{\psi})$ requires solving the integral in Equation (1). While this might appear daunting, such integrals can often be efficiently approximated numerically using Clenshaw-Curtis quadrature. The better known trapezoidal rule, which corresponds to the two-point Newton-Cotes quadrature rule, has an exponential convergence when the integrand is periodic and the range of integration corresponds to its period. Clenshaw-Curtis quadrature takes advantage of this property by using a change of variables followed by a cosine transform. This extends the exponential convergence of the trapezoidal rule for periodic functions to any Lipschitz continuous function. As a result, the number of evaluation points required to reach convergence grows with the Lipschitz constant of the function.

**Backward integration** Training the integrand neural network $f$ requires evaluating the gradient of $F$ with respect to its parameters. While this gradient could be obtained by backpropagating directly through the integral solver, this would also result in a memory footprint that grows linearly with the number of integration steps. Instead, the derivative of an integral with respect to a parameter $\omega$ can be expressed with the Leibniz integral rule:

$$\frac{\mathrm{d}}{\mathrm{d}\omega} \left( \int_{a(\omega)}^{b(\omega)} f(t; \omega) \, \mathrm{d}t \right) = f(b(\omega); \omega) \frac{\mathrm{d}}{\mathrm{d}\omega} b(\omega) - f(a(\omega); \omega) \frac{\mathrm{d}}{\mathrm{d}\omega} a(\omega) + \int_{a(\omega)}^{b(\omega)} \frac{\partial}{\partial \omega} f(t; \omega) \, \mathrm{d}t. \tag{2}$$

Applying Equation (2) to evaluate the derivative of Equation (1) with respect to the parameters $\boldsymbol{\psi}$, we find

$$\nabla_{\boldsymbol{\psi}} F(x; \boldsymbol{\psi}) = f(x; \boldsymbol{\psi}) \nabla_{\boldsymbol{\psi}} (x) - f(0; \boldsymbol{\psi}) \nabla_{\boldsymbol{\psi}} (0) + \int_0^x \nabla_{\boldsymbol{\psi}} f(t; \boldsymbol{\psi}) \, \mathrm{d}t + \nabla_{\boldsymbol{\psi}} \beta$$

$$= \int_0^x \nabla_{\boldsymbol{\psi}} f(t; \boldsymbol{\psi}) \, \mathrm{d}t + \nabla_{\boldsymbol{\psi}} \beta. \tag{3}$$

When using a UMNN block in a neural architecture, it is also important to be able to compute its derivative with respect to its input $x$. In this case, applying Equation (2) leads to

$$\frac{\mathrm{d}}{\mathrm{d}x} F(x; \boldsymbol{\psi}) = f(x; \boldsymbol{\psi}). \tag{4}$$

Equations ([3]) and ([4]) make the memory footprint for the backward pass independent from the number of integration steps, and therefore also from the desired accuracy. Indeed, instead of computing the gradient of the integral (which requires keeping track of all the integration steps), we integrate the gradient (which is memory efficient, as this corresponds to summing gradients at different evaluation points). We provide the pseudo-code of the forward and backward passes using Clenshaw-Curtis quadrature in Appendix [B].

**Numerical inversion** In UMMNs, the modeled monotonic function $F$ is arbitrary. As a result, computing its inverse cannot be done analytically. However, since $F$ is strictly monotonic, it admits a unique inverse $x$ for any point $y = F(x; \boldsymbol{\psi})$ in its image, therefore inversion can be computed efficiently with common root-finding algorithms. In our experiments, search algorithms such as the bisection method proved to be fast enough.

# 3 UMNN autoregressive models

## 3.1 Normalizing flows

A Normalizing Flow [NF, Rezende and Mohamed, 2015] is defined as a sequence of invertible transformations $\boldsymbol{u}_i : \mathbb{R}^d \to \mathbb{R}^d$ $(i = 1, ..., k)$ composed together to create an expressive invertible mapping $\boldsymbol{u} = \boldsymbol{u}_1 \circ \cdots \circ \boldsymbol{u}_k : \mathbb{R}^d \to \mathbb{R}^d$. It is common for normalizing flows to stack the same parametric function $\boldsymbol{u}_i$ (with different parameters values) and to reverse variables ordering after each transformation. For this reason, we will focus on how to build one of these repeated transformations, which we further refer to as $\boldsymbol{g} : \mathbb{R}^d \to \mathbb{R}^d$.

**Density estimation** NFs are most commonly used for density estimation, that map empirical samples to unstructured noise. Using normalizing flows, we define a bijective mapping $\boldsymbol{u}(\cdot; \boldsymbol{\theta}) : \mathbb{R}^d \to \mathbb{R}^d$ from a sample $\boldsymbol{x} \in \mathbb{R}^d$ to a latent vector $\boldsymbol{z} \in \mathbb{R}^d$ equipped with a density $p_Z(\boldsymbol{z})$. The transformation $\boldsymbol{u}$ implicitly defines a density $p(\boldsymbol{x}; \boldsymbol{\theta})$ as given by the change of variables formula,

$$p(\boldsymbol{x}; \boldsymbol{\theta}) = p_Z(\boldsymbol{u}(\boldsymbol{x}; \boldsymbol{\theta})) \left| \det J_{\boldsymbol{u}(\boldsymbol{x}; \boldsymbol{\theta})} \right|, \tag{5}$$

where $J_{\boldsymbol{u}(\boldsymbol{x}; \boldsymbol{\theta})}$ is the Jacobian of $\boldsymbol{u}(\boldsymbol{x}; \boldsymbol{\theta})$ with respect to $\boldsymbol{x}$. The resulting model is trained by maximizing the likelihood of the data $\{\boldsymbol{x}^1, ..., \boldsymbol{x}^N\}$.

**Variational auto-encoders** NFs are also used in VAE to improve posterior modeling. In this case, a normalizing flow transforms a distribution $p_Z$ into a complex distribution $q$ which can better model the variational posterior. The change of variables formula yields

$$q(\boldsymbol{u}(\boldsymbol{z}; \boldsymbol{\theta})) = p_Z(\boldsymbol{z}) \left| \det J_{\boldsymbol{u}(\boldsymbol{z}; \boldsymbol{\theta})} \right|^{-1}. \tag{6}$$

## 3.2 Autoregressive transformations

To be of practical use, NFs must be composed of transformations for which the determinant of the Jacobian can be computed efficiently, otherwise its evaluation would be running in $\mathcal{O}(d^3)$. A common solution consists in making the transformation $\boldsymbol{g}$ autoregressive, i.e., such that $\boldsymbol{g}(\boldsymbol{x}; \boldsymbol{\theta})$ can be rewritten as a vector of $d$ scalar functions,

$$\boldsymbol{g}(\boldsymbol{x}; \boldsymbol{\theta}) = \begin{bmatrix} g^1(x_1; \boldsymbol{\theta}) & \cdots & g^i(\boldsymbol{x}_{1:i}; \boldsymbol{\theta}) & \cdots & g^d(\boldsymbol{x}_{1:d}; \boldsymbol{\theta}) \end{bmatrix},$$

where $\boldsymbol{x}_{1:i} = \begin{bmatrix} x_1 & \cdots & x_i \end{bmatrix}^T$ is the vector including the $i$ first elements of the full vector $\boldsymbol{x}$. The Jacobian of this function is lower triangular, which makes the computation of its determinant $\mathcal{O}(d)$. Enforcing the bijectivity of each component $g^i$ is then sufficient to make $\boldsymbol{g}$ bijective as well.

For the multivariate density $p(\boldsymbol{x}; \boldsymbol{\theta})$ induced by $\boldsymbol{g}(\boldsymbol{x}; \boldsymbol{\theta})$ and $p_Z(\boldsymbol{z})$, we can use the chain rule to express the joint probability of $\boldsymbol{x}$ as a product of $d$ univariate conditional densities,

$$p(\boldsymbol{x}; \boldsymbol{\theta}) = p(x_1; \boldsymbol{\theta}) \prod_{i=1}^{d-1} p(x_{i+1} | \boldsymbol{x}_{1:i}; \boldsymbol{\theta}). \tag{7}$$

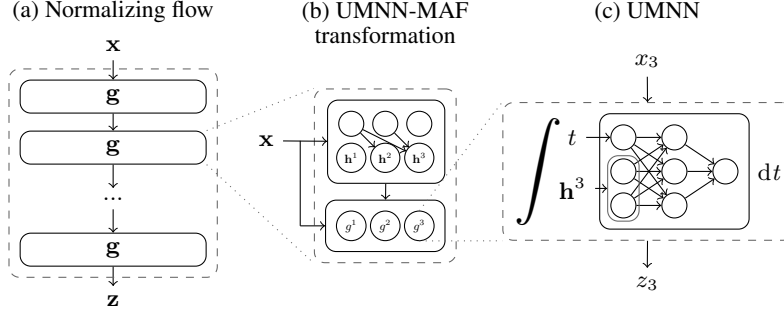

Figure 1: **(a)** A normalizing flow made of repeated UMNN-MAF transformations $\boldsymbol{g}$ with identical architectures. **(b)** A UMNN-MAF which transforms a vector $\boldsymbol{x} \in \mathbb{R}^3$. **(c)** The UMNN network used to map $x_3$ to $z_3$ conditioned on the embedding $\boldsymbol{h}^3(\boldsymbol{x}_{1:2})$.

When $p_Z(\boldsymbol{z})$ is a factored distribution $p_Z(\boldsymbol{z}) = \prod_{i=1}^d p(z_i)$, we identify that each component $z_i$ coupled with the corresponding function $g^i$ encodes for the conditional $p(x_i|\boldsymbol{x}_{1:i-1}; \boldsymbol{\theta})$. Autoregressive transformations strongly rely on the expressiveness of the scalar functions $g^i$. In this work, we propose to use UMNNs to create powerful bijective scalar transformations.

### 3.3 UMNN autoregressive transformations (UMNN-MAF)

We now combine UMNNs with an embedding of the conditioning variables to build invertible autoregressive functions $g^i$. Specifically, we define

$$
\begin{aligned}
g^i(\boldsymbol{x}_{1:i}; \boldsymbol{\theta}) &= F^i(x_i, \boldsymbol{h}^i(\boldsymbol{x}_{1:i-1}; \boldsymbol{\phi}^i); \boldsymbol{\psi}^i) \\
&= \int_0^{x_i} f^i(t, \boldsymbol{h}^i(\boldsymbol{x}_{1:i-1}; \boldsymbol{\phi}^i); \boldsymbol{\psi}^i) \, \mathrm{d}t + \beta^i(\boldsymbol{h}^i(\boldsymbol{x}_{1:i-1}; \boldsymbol{\phi}^i)),
\end{aligned} \tag{8}
$$

where $\boldsymbol{h}^i(\cdot; \boldsymbol{\phi}^i): \mathbb{R}^{i-1} \to \mathbb{R}^q$ is a $q$-dimensional neural embedding of the conditioning variables $\boldsymbol{x}_{1:i-1}$ and $\beta(\cdot)^i: \mathbb{R}^{i-1} \to \mathbb{R}$. Both degenerate into constants for $g^1(\boldsymbol{x}_1)$. The parameters $\boldsymbol{\theta}$ of the whole transformation $\boldsymbol{g}(\cdot; \boldsymbol{\theta})$ is the union of all parameters $\boldsymbol{\phi}^i$ and $\boldsymbol{\psi}^i$. For simplicity we remove the parameters of the networks by rewriting $f^i(\cdot; \boldsymbol{\psi}^i)$ as $f^i(\cdot)$ and $\boldsymbol{h}^i(\cdot; \boldsymbol{\phi}^i)$ as $\boldsymbol{h}^i(\cdot)$.

In our implementation, we use a Masked Autoregressive Network [Germain et al., 2015, Kingma et al., 2016, Papamakarios et al., 2017] to simultaneously parameterize the $d$ embeddings. In what follows we refer to the resulting UMNN autoregressive transformation as UMNN-MAF. Figure 1 summarizes the complete architecture.

**Log-density** The change of variables formula applied to the UMNN autoregressive transformation results in the log-density

$$
\begin{aligned}
\log p(\boldsymbol{x}; \boldsymbol{\theta}) &= \log p_Z(\boldsymbol{g}(\boldsymbol{x}; \boldsymbol{\theta})) \left|\det J_{\boldsymbol{g}(\boldsymbol{x}; \boldsymbol{\theta})}\right| \\
&= \log p_Z(\boldsymbol{g}(\boldsymbol{x}; \boldsymbol{\theta})) + \log \left| \prod_{i=1}^d \frac{\partial F^i(x_i, \boldsymbol{h}^i(\boldsymbol{x}_{1:i-1}))}{\partial x_i} \right| \\
&= \log p_Z(\boldsymbol{g}(\boldsymbol{x}; \boldsymbol{\theta})) + \sum_{i=1}^d \log f^i(x_i, \boldsymbol{h}^i(\boldsymbol{x}_{1:i-1})).
\end{aligned} \tag{9}
$$

Therefore, the transformation leads to a simple expression of (the determinant of) its Jacobian, which can be computed efficiently with a single forward pass. This is different from FFJORD [Grathwohl et al., 2018] which relies on numerical methods to compute both the Jacobian and the transformation between the data and the latent space. Therefore our proposed method makes the computation of the Jacobian exact and efficient at the same time.

**Sampling** Generating samples require evaluating the inverse transformation $\boldsymbol{g}^{-1}(\boldsymbol{z}; \boldsymbol{\theta})$. The components of the inverse vector $\boldsymbol{x}^{\text{inv}} = \boldsymbol{g}^{-1}(\boldsymbol{z}; \boldsymbol{\theta})$ can be computed recursively by inverting each

component of $\boldsymbol{g}(\boldsymbol{x}; \boldsymbol{\theta})$:

$$x_1^{\mathrm{inv}} = \left(g^1\right)^{-1} \left(z_1; \boldsymbol{h}^1\right) \qquad \text{if} \quad i = 1 \qquad (10)$$

$$x_i^{\mathrm{inv}} = \left(g^i\right)^{-1} \left(z_i; \boldsymbol{h}^i \left(\boldsymbol{x}_{1:i-1}^{\mathrm{inv}}\right)\right) \qquad \text{if} \quad i > 1 \qquad (11)$$

where $(g^i)^{-1}$ is the inverse of $g^i$. Another approach to invert an autoregressive model would be to approximate its inverse with another autoregressive network [Oord et al., 2018]. In this case, the evaluation of the approximated inverse model is as fast as the forward model.

**Universality**  Since the proof is straightforward, we only sketch that UMNN-MAF is a universal density approximator of continuous random variables. We rely on the inverse sampling theorem to prove that UMNNs are universal approximators of continuously derivable ($\mathbb{C}^1$) monotonic functions. Indeed, if UMNNs can represent any $\mathbb{C}^1$ monotonic function, then they can also represent the (inverse) cumulative distribution function of any continuous random variable. Any continuously derivable function $f : \mathcal{D} \to \mathcal{I}$ can be expressed as the following integral: $f(x) = \int_a^x \frac{df}{dx} dx + f(a), \quad \forall x, a \in \mathcal{D}$. The derivative $\frac{df}{dx}$ is a continuous positive function and the universal approximation theorem of NNs ensures it can be successfully approximated with a NN of sufficient capacity (such as those used in UMNNs).

## 4   Related work

The most similar work to UMNN-MAF are certainly Neural Autoregressive Flow [NAF, Huang et al., 2018] and Block Neural Autoregressive Flow [B-NAF, De Cao et al., 2019], which both rely on strictly monotonic transformations for building bijective mappings. In NAF, transformations are defined as neural networks which activation functions are all constrained to be strictly monotonic and which weights are the output of a strictly positive and autoregressive HyperNetwork [Ha et al., 2017]. Huang et al. [2018] shows that NAFs are universal density approximators. In B-NAF, the authors improve on the scalability of the NAF architecture by making use of masking operations instead of HyperNetworks. They also present a proof of the universality of B-NAF, which extends to UMNN-MAF. Our work differs from both NAF and B-NAF in the sense that the UMNN monotonic transformation is based on free-form neural networks for which no constraint, beyond positiveness of the output, is enforced on the hypothesis class. This leads to multiple advantages: it enables the use of any state-of-the-art neural architecture, simplifies weight initialization, and leads to a more lightweight evaluation of the Jacobian.

More generally, UMNN-MAF relates to works on normalizing flows built upon autoregressive networks and affine transformations. Germain et al. [2015] first introduced masking as an efficient way to build autoregressive networks, and proposed autoregressive networks for density estimation of high dimensional binary data. Masked Autoregressive Flows [Papamakarios et al., 2017] and Inverse Autoregressive Flows [Kingma et al., 2016] have generalized this approach to real data, respectively for density estimation and for latent posterior representation in variational auto-encoders. More recently, Oliva et al. [2018] proposed to stack various autoregressive architectures to create powerful reversible transformations. Meanwhile, Jaini et al. [2019] proposed a new Sum-of-Squares flow that is defined as the integral of a second order polynomial parametrized by an autoregressive NN.

With NICE, Dinh et al. [2015] introduced coupling layers, which correspond to bijective transformations splitting the input vector into two parts. They are defined as

$$\boldsymbol{z}_{1:k} = \boldsymbol{x}_{1:k} \quad \text{and} \quad \boldsymbol{z}_{k+1:d} = e^{\boldsymbol{\sigma}(\boldsymbol{x}_{1:k})} \odot \boldsymbol{x}_{k+1:d} + \boldsymbol{\mu}(\boldsymbol{x}_{1:k}), \qquad (12)$$

where $\boldsymbol{\sigma}$ and $\boldsymbol{\mu}$ are two unconstrained functions $\mathbb{R}^{d-k} \to \mathbb{R}^{d-k}$. The same authors introduced RealNVP [Dinh et al., 2017], which combines coupling layers with normalizing flows and multi-scale architectures for image generation. Glow [Kingma and Dhariwal, 2018] extends RealNVP by introducing invertible 1x1 convolutions between each step of the flow. In this work we have used UMNNs in the context of autoregressive architectures, however UMNNs could also be applied to replace the linear transformation in coupling layers.

Finally, our architecture also shares a connection with Neural Ordinary Differential Equations [NODE, Chen et al., 2018]. The core idea of this architecture is to learn an ordinary differential equation which dynamic is parameterized by a neural network. Training can be carried

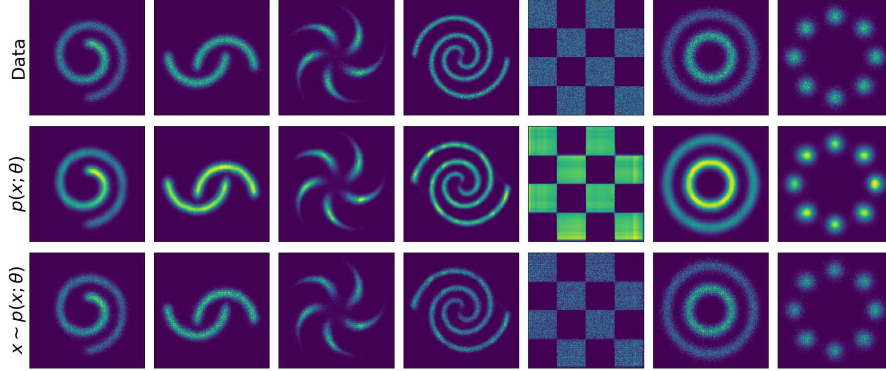

Figure 2: Density estimation and sampling with a UMNN-MAF network on 2D toy problems. **Top**: Samples from the empirical distribution $p(\boldsymbol{x})$. **Middle**: Learned density $p(\boldsymbol{x};\boldsymbol{\theta})$. **Bottom**: Samples drawn by numerical inversion. UMNN-MAF manages to precisely capture multi-modal and/or discontinuous distributions. Sampling is possible even if the model is not invertible analytically.

out by backpropagating efficiently through the ODE solver, with constant memory requirements. Among other applications, NODE can be used to model a continuous normalizing flow with a free-form Jacobian as in FFJORD [Grathwohl et al., 2018]. Similarly, a UMNN transformation can be seen as a structured neural ordinary differential equation in which the dynamic of the vector field is separable and can be solved efficiently by direct integration.

## 5 Experiments

In this section, we evaluate the expressiveness of UMNN-MAF on a variety of density estimation benchmarks, as well as for approximate inference in variational auto-encoders. The source code to reproduce our experiments will be made available on Github at the end of the reviewing process.

Experiments were carried out using the same integrand neural network in the UMNN component – i.e., in Equation 8, $f^i = f$ with shared weights $\boldsymbol{\psi}^i = \boldsymbol{\psi}$ for $i \in \{1, \dots, d\}$. The functions $\beta^i$ are taken to be equal to one of the outputs of the embedding network. We observed in our experiments that sharing the same integrand function does not impact performance. Therefore, the neural embedding function $\boldsymbol{h}^i$ must produce a fixed size output for $i \in \{1, \dots, d\}$.

### 5.1 2D toy problems

We first train a UMNN-MAF on 2-dimensional toy distributions, as defined by Grathwohl et al. [2018]. To train the model, we minimize the negative log-likelihood of observed data

$$L(\boldsymbol{\theta}) = -\sum_{n=1}^{N} \left[ \log p_Z(\boldsymbol{g}(\boldsymbol{x}^n; \boldsymbol{\theta})) + \sum_{i=1}^{d} \log f(x_i^n, \boldsymbol{h}^i(\boldsymbol{x}_{1:i-1}^n)) \right]. \tag{13}$$

The flow used to solve these tasks is the same for all distributions and is composed of a single transformation. More details can be found in Appendix A.1.

Figure 2 demonstrates that our model is able to learn a change of variables that warps a simple isotropic Gaussian into multimodal and/or discontinuous distributions. We observe from the figure that our model precisely captures the density of the data. We also observe that numerical inversion for generating samples yields good results.

### 5.2 Density estimation

We further validate UMNN-MAF by comparing it to state-of-the-art normalizing flows. We carry out experiments on tabular datasets (POWER, GAS, HEPMASS, MINIBOONE, BSDS300) as well as on MNIST. We follow the experimental protocol of Papamakarios et al. [2017]. All training hyper-parameters and architectural details are given in Appendix A.1. For each dataset, we report

Table 1: Average negative log-likelihood on test data over 3 runs, error bars are equal to the standard deviation. Results are reported in nats for tabular data and bits/dim for MNIST; lower is better. The best performing architecture for each dataset is written in bold and the best performing architecture per category is underlined. (a) Non-autoregressive models, (b) Autoregressive models, (c) Monotonic and autoregressive models. UMNN outperforms other monotonic transformations on 4 tasks over 6 and is the overall best performing model on 2 tasks over 6.

| | Dataset | **POWER** | **GAS** | **HEPMASS** | **MINIBOONE** | **BSDS300** | **MNIST** |
|---|---|---|---|---|---|---|---|
| | RealNVP - Dinh et al. [2017] | $-0.17_{\pm.01}$ | $-8.33_{\pm.14}$ | $18.71_{\pm.02}$ | $13.55_{\pm.49}$ | $-153.28_{\pm1.78}$ | - |
| (a) | Glow - Kingma and Dhariwal [2018] | $-0.17_{\pm.01}$ | $-8.15_{\pm.40}$ | $19.92_{\pm.08}$ | $11.35_{\pm.07}$ | $-155.07_{\pm.03}$ | - |
| | FFJORD - Grathwohl et al. [2018] | $\underline{-0.46}_{\pm.01}$ | $\underline{-8.59}_{\pm.12}$ | $\underline{14.92}_{\pm.08}$ | $\underline{10.43}_{\pm.04}$ | $\underline{-157.40}_{\pm.19}$ | - |
| | MADE - Germain et al. [2015] | $3.08_{\pm.03}$ | $-3.56_{\pm.04}$ | $20.98_{\pm.02}$ | $15.59_{\pm.50}$ | $-148.85_{\pm.28}$ | $2.04_{\pm.01}$ |
| (b) | MAF - Papamakarios et al. [2017] | $-0.24_{\pm.01}$ | $-10.08_{\pm.02}$ | $17.70_{\pm.02}$ | $11.75_{\pm.44}$ | $-155.69_{\pm.28}$ | $1.89_{\pm.01}$ |
| | TAN - Oliva et al. [2018] | $\underline{-0.60}_{\pm.01}$ | $\mathbf{-12.06}_{\pm.02}$ | $\mathbf{13.78}_{\pm.02}$ | $\underline{11.01}_{\pm.48}$ | $\mathbf{-159.80}_{\pm.07}$ | $\underline{1.19}$ |
| | NAF - Huang et al. [2018] | $-0.62_{\pm.01}$ | $-11.96_{\pm.33}$ | $15.09_{\pm.40}$ | $\mathbf{8.86}_{\pm.15}$ | $-157.73_{\pm.30}$ | - |
| (c) | B-NAF - De Cao et al. [2019] | $-0.61_{\pm.01}$ | $\underline{-12.06}_{\pm.09}$ | $14.71_{\pm.38}$ | $8.95_{\pm.07}$ | $-157.36_{\pm.03}$ | - |
| | SOS - Jaini et al. [2019] | $-0.60_{\pm.01}$ | $-11.99_{\pm.41}$ | $15.15_{\pm.1}$ | $8.90_{\pm.11}$ | $-157.48_{\pm.41}$ | $1.81$ |
| | **UMNN-MAF (ours)** | $\mathbf{-0.63}_{\pm.01}$ | $-10.89_{\pm.7}$ | $\underline{13.99}_{\pm.21}$ | $9.67_{\pm.13}$ | $\underline{-157.98}_{\pm.01}$ | $\mathbf{1.13}_{\pm.02}$ |

results on test data for our best performing model (selected on the validation data). At testing time we use a large number of integration steps (100) to compute the integral, this ensures its correctness and avoids misestimating the performance of UMNN-MAF.

Table 1 summarizes our results, where we can see that on tabular datasets, our method is competitive with other normalizing flows. For POWER, our architecture slightly outperforms all others. It is also better than other monotonic networks (category (c)) on 3 tabular datasets over 5. From these results, we could conclude that Transformation Autoregressive Networks [TAN, Oliva et al., 2018] is overall the best method for density estimation. It is however important to note that TAN is a flow composed of many heterogeneous transformations (both autoregressive and non-autoregressive). For this reason, it should not be directly compared to the other models which respective results are specific to a single architecture. However, TAN provides the interesting insight that combining heterogeneous components into a flow leads to better results than an homogeneous flow.

Notably, we do not make use of a multi-scale architecture to train our model on MNIST. On this task, UMNN-MAF slightly outperforms all other models by a reasonable margin. Samples generated by a conditional model are shown on Figure 3, for which it is worth noting that UMNN-MAF is the first monotonic architecture that has been inverted to generate samples. Indeed, MNIST can be considered as a high dimensional dataset ($d = 784$) for standard feed forward neural networks which autoregressive networks are part of. NAF and B-NAF do not report any result for this benchmark, presumably because of memory explosion. In comparison, BSDS300, which data dimension is one order of magnitude smaller than MNIST ($63 \ll 784$), are the largest data they have tested on. Table 2 shows the number of parameters used by UMNN-MAF in comparison to B-NAF and NAF. For bigger datasets, UMNN-MAF requires less parameters than NAF to reach similar or better performance. This could explain why NAF has never been used for density estimation on MNIST.

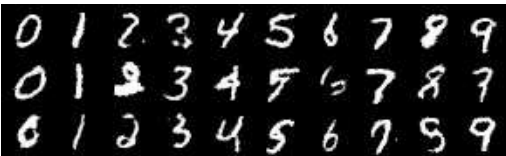

Figure 3: Samples generated by numerical inversion of a conditional UMNN-MAF trained on MNIST. Samples $\mathbf{z}$ are drawn from an isotropic Gaussian with $\sigma = .75$. See Appendix C for more details.

Table 2: Comparison of the number of parameters between NAF, B-NAF and UMNN-MAF. In high dimensional datasets, UMNN-MAF requires fewer parameters than NAF and a similar number to B-NAF.

| Dataset | NAF | B-NAF | UMNN-MAF |
|---|---|---|---|
| **POWER** ($d = 6$) | 4.14e5 | 3.07e5 | 5.09e5 |
| **GAS** ($d = 8$) | 4.02e5 | 5.44e5 | 8.15e5 |
| **HEPMASS** ($d = 21$) | 9.27e6 | 3.72e6 | 3.62e6 |
| **MINIBOONE** ($d = 43$) | 7.49e6 | 4.09e6 | 3.46e6 |
| **BSDS300** ($d = 63$) | 3.68e7 | 8.76e6 | 1.56e7 |

Table 3: Average negative evidence lower bound of VAEs over 3 runs, error bars are equal to the standard deviation. Results are reported in bits per dim for Freyfaces and in nats for the other datasets; lower is better. UMNN-NAF is performing slightly better than IAF but is outperformed by B-NAF. We believe that the gap in performance between B-NAF and UMNN is due to the way the NF is conditioned by the encoder's output.

|  | Dataset | MNIST | Freyfaces | Omniglot | Caltech 101 |
|---|---|---|---|---|---|
| (a) | VAE - Kingma and Welling [2013] | $86.65_{\pm.06}$ | $4.53_{\pm.02}$ | $104.28_{\pm.39}$ | $110.80_{\pm.46}$ |
| | Planar - Rezende and Mohamed [2015] | $86.06_{\pm.32}$ | $4.40_{\pm.06}$ | $102.65_{\pm.42}$ | $109.66_{\pm.42}$ |
| | IAF - Kingma et al. [2016] | $84.20_{\pm.17}$ | $4.47_{\pm.05}$ | $102.41_{\pm.04}$ | $111.58_{\pm.38}$ |
| | Sylvester - Berg et al. [2018] | $83.32_{\pm.06}$ | $4.45_{\pm.04}$ | $99.00_{\pm.04}$ | $104.62_{\pm.29}$ |
| | FFJORD - Grathwohl et al. [2018] | $82.82_{\pm.01}$ | $4.39_{\pm.01}$ | $98.33_{\pm.09}$ | $104.03_{\pm.43}$ |
| (b) | B-NAF - De Cao et al. [2019] | $83.59_{\pm.15}$ | $4.42_{\pm.05}$ | $100.08_{\pm.07}$ | $105.42_{\pm.49}$ |
| | **UMNN-MAF** (ours) | $84.11_{\pm.05}$ | $4.51_{\pm.01}$ | $100.98_{\pm.13}$ | $110.45_{\pm.69}$ |

## 5.3 Variational auto-encoders

To assess the performance of our model, we follow the experimental setting of Berg et al. [2018] for VAE. The encoder and the decoder architectures can be found in the appendix of their paper. In VAE it is usual to let the encoder output the parameters of the flow. For UMNN-MAF, this would cause the encoder output's dimension to be too large. Instead, the encoder output is passed as additional entries of the UMNN-MAF. Like other architectures, the UMNN-MAF also takes as input a vector of noise drawn from an isotropic Gaussian of dimension 64.

Table 3 presents our results. It shows that on MNIST and Omniglot, UMNN-MAF slightly outperforms the classical VAE as well as planar flows. Moreover, on these datasets and Freyfaces, IAF, B-NAF and UMNN-MAF achieve similar results. FFJORD is the best among all, however it is worth noting that the roles of encoder outputs in FFJORD, B-NAF, IAF and Sylvester are all different. We believe that the heterogeneity of the results could be, at least in part, due to the different amortizations.

## 6 Discussion and summary

**Static integral quadrature can be inaccurate.** Computing the integral with static Clenshaw-Curtis quadrature only requires the evaluation of the integrand at predefined points. As such, batches of points can be processed all at once, which makes static Clenshaw-Curtis quadrature well suited for neural networks. However, static quadratures do not account for the error made during the integration. As a consequence, the quadrature is inaccurate when the integrand is not smooth enough and the number of integration steps is too small. In this work, we have reduced the integration error by applying the normalization described by Gouk et al. [2018] in order to control the Lipschitz constant of the integrand and appropriately set the number of integration steps. We observed that as long as the Lipschitz constant of the network does not increase dramatically ($< 1000$), a reasonable number of integration steps ($< 100$) is sufficient to ensure the convergence of the quadrature. An alternative solution would be to use dynamic quadrature such as dynamic Clenshaw-Curtis.

**Efficiency of numerical inversion.** Architectures relying on linear transformations [Papamakarios et al., 2017, Kingma et al., 2016, Dinh et al., 2017, Kingma and Dhariwal, 2018] are trivially exactly and efficiently invertible. In contrast, the UMNN transformation has no analytic inverse. Nevertheless, it can be inverted numerically using root-finding algorithms. Since most such algorithms rely on multiple nested evaluations of the function to be inverted, applying them naively to a numerical integral would quickly become very inefficient. However, the Clenshaw-Curtis quadrature is part of the nested quadrature family, meaning that the evaluation of the integral at multiple nested points can take advantage of previous evaluations and thus be implemented efficiently. As an alternative, Oord et al. [2018] have shown that an invertible model can always be distilled to learn its inverse, and thus make the inversion efficient whatever the cost of inversion of the original model.

**Scalability and complexity analysis.** UMNN-MAF is particularly well suited for density estimation because the computation of the Jacobian only requires a single forward evaluation of a NN.

Together with the Leibniz integral rule, they make the evaluation of the log-likelihood derivative as memory efficient as usual supervised learning, which is equivalent to a single backward pass on the computation graph. By contrast, density estimation with previous monotonic transformations typically requires a backward evaluation of the computation graph of the transformer NN to obtain the Jacobian. Then, this pass must be evaluated backward again in order to obtain the log-likelihood derivative. Both NAF and B-NAF provide a method to make this computation numerically stable, however both fail at not increasing the size of the computation graph of the log-likelihood derivative, hence leading to a memory overhead. The memory saved by the Leibniz rule may serve to speed up the quadrature computation. In the case of static Clenshaw-Curtis, the function values at each evaluation point can be computed in parallel using batch of points. In consequence, when the GPU memory is large enough to store "meta-batches" of size $d \times N \times B$ (with $d$ the dimension of the data, $N$ the number of integration steps and $B$ the batch size) the computation is approximately as fast as a forward evaluation of the integrand network.

**Summary**  We have introduced Unconstrained Monotonic Neural Networks, a new invertible transformation built upon free-form neural networks allowing the use of any state-of-the-art architecture. Monotonicity is guaranteed without imposing constraints on the expressiveness of the hypothesis class, contrary to classical approaches. We have shown that the resulting integrated neural network can be evaluated efficiently using standard quadrature rule while its inverse can be computed using numerical algorithms. We have shown that our transformation can be composed into an autoregressive flow, with competitive or state-of-the-art results on density estimation and variational inference benchmarks. Moreover, UMNN is the first monotonic transformation that has been successfully applied for density estimation on high dimensional data distributions (MNIST), showing better results than the classical approaches.

We identify several avenues for improvement and further research. First, we believe that numerical integration could be fasten up during training, by leveraging the fact that controlled numerical errors can actually help generalization. Moreover, the UMNN transformation would certainly profit from using a dynamic integration scheme, both in terms of accuracy and efficiency. Second, it would be worth comparing the newly introduced monotonic transformation with common approaches for modelling monotonic functions in machine learning. On a similar track, these common approaches could be combined into an autoregressive flow as shown in Section 3.3. Finally, our monotonic transformation could be used within other neural architectures than generative autoregressive networks, such as multi-scale architectures [Dinh et al., 2017] and learnable 1D convolutions [Kingma and Dhariwal, 2018].

### Acknowledgments

The authors would like to acknowledge Matthia Sabatelli, Nicolas Vecoven, Antonio Sutera and Louis Wehenkel for useful feedback on the manuscript. They would also like to thank the anonymous reviewers for many relevant remarks. Antoine Wehenkel is a research fellow of the F.R.S.-FNRS (Belgium) and acknowledges its financial support.

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
