[Supplementary Material]

# A   Experimental setup

## A.1   Density estimation and toy problems hyperparameters

Table 4 reports the training configurations for the 2D toy problems and the 5 tabular datasets. For tabular data the best performing architecture has been found after some preliminary experiments, while this was not needed for the 2D toy problems. During our preliminary experiments we tested different integrand network architectures, we tested on the number of hidden layers $L \in \{3, 4\}$ and on their dimension $D \in \{50, 100, 150, 200\}$. The architecture of the embedding networks is the best performing MADE network used in NAF [Huang et al., 2018]. We used the Adam optimizer and tried different learning rate $\lambda \in \{10^{-3}, 5 \times 10^{-4}, 10^{-4}\}$. When the learning rate chosen was greater than $10^{-4}$ we schedule once the learning rate to $10^{-4}$ after the first plateau. We also tested for different weights decay values $W \in \{10^{-5}, 10^{-2}\}$. The batch size was chosen to be as big as possible while not harming the learning procedure. We observed during our preliminary experiments that choosing the number of integration steps at random (uniformly from 20 to 100) for each batch regularizes the complexity of the integral. For MNIST, we observed that 25 integration steps was enough if the Lipschitz constant of the network is constraint (with the normalization proposed by Gouk et al. [2018]) to be smaller than 1.5.

| Dataset | POWER | GAS | HEPMASS | MINIBOONE | BSDS300 | MNIST | 2D Toys |
|---|---|---|---|---|---|---|---|
| Lipschitz | - | - | - | - | 2.5 | 1.5 | - |
| N°integ. steps | rand | rand | rand | rand | rand | 25 | 50 |
| Embedding net | $2 \times 100$ | $2 \times 100$ | $2 \times 512$ | $1 \times 512$ | $2 \times 1024$ | $1 \times 1024$ | $4 \times 50$ |
| Integrand net ($L \times D$) | $4 \times 150$ | $3 \times 200$ | $4 \times 200$ | $3 \times 50$ | $4 \times 150$ | $3 \times 150$ | $4 \times 50$ |
| Learning rate ($\lambda$) | $10^{-3}$ | $10^{-3}$ | $10^{-3}$ | $10^{-3}$ | $10^{-4}$ | $10^{-3}$ | $10^{-3}$ |
| N°flows | 5 | 10 | 5 | 3 | 5 | 5 | 1 |
| Embedding Size | 30 | 30 | 30 | 30 | 30 | 30 | 10 |
| Weight decay ($W$) | $10^{-5}$ | $10^{-2}$ | $10^{-4}$ | $10^{-2}$ | $10^{-2}$ | $10^{-2}$ | $10^{-5}$ |
| Batch size | 10000 | 10000 | 100 | 500 | 100 | 100 | 100 |

Table 4: Training configurations for density estimation and toy problems.

## A.2   Variational auto-encoders

Table 5 presents the architectural settings of the normalizing flows used inside the variational auto-encoders. The number of values outputted by the encoder is always taken to be equal to $320$. These values as well as the $64$-dimensional noise vector are the inputs of the embedding network which is constantly made of one hidden layer of $1280$ neurons. We have performed a small grid search on the integrand network architecture, we took a look at 2 different number $L \in \{3, 4\}$ of hidden layers of dimensions $D \in \{100, 150\}$.

| Dataset | MNIST | Freyfaces | Omniglot | Caltech 101 |
|---|---|---|---|---|
| Lipschitz | - | - | - | - |
| N°integ. steps | rand | rand | rand | rand |
| Encoder Output | 320 | 320 | 320 | 320 |
| Embedding net | $1 \times 1280$ | $1 \times 1280$ | $1 \times 1280$ | $1 \times 1280$ |
| Integrand net | $4 \times 100$ | $3 \times 100$ | $4 \times 100$ | $4 \times 100$ |
| N°flows | 16 | 8 | 16 | 16 |
| Embedding Size | 30 | 30 | 30 | 30 |

Table 5: Training configurations of variational auto-encoder.

 # B Clenshaw-Curtis module

---

**Algorithm 1** Clenshaw-Curtis quadrature

---

| | |
|---|---|
| *Input:* | $x$: A tensor of scalar values that represent the superior integration bounds. |
| | $\boldsymbol{h}$: A tensor of vectors that representing embeddings. |
| *Output:* | $F$: A tensor of scalar values that represent the integral of $\int_0^x f(t; \boldsymbol{h})\, \mathrm{d}t$ . |
| *Hyper-parameters:* | $f$: A derivable function $\mathbb{R} \to \mathbb{R}$. |
| | $N$: The number of integration steps. |

1: **procedure** FORWARD($x$, $\boldsymbol{h}$; $f$, $N$)
2:     ▷ Compute weights and evaluation steps for Clenshaw-Curtis quadrature
3:     $\boldsymbol{w}, \boldsymbol{\delta}_x = $ COMPUTE_CC_WEIGHTS($N$)
4:     $F = 0$
5:     **for** $i \in [1, N]$ **do**
6:         $x_i = x_0 + \frac{1}{2}(x - x_0)(\boldsymbol{\delta}_x[i] + 1)$                    ▷ Compute the next point to evaluate
7:         $\delta_F = f(x_i; \boldsymbol{h})$
8:         $F = F + \boldsymbol{w}[i]\delta_F$
9:     **end for**
10:     $F = \frac{F}{2}(x - x_0)$
11:     **return** $F$
12: **end procedure**

| | |
|---|---|
| *Inputs:* | $x$: A tensor of scalar values that represent the superior integration bounds. |
| | $\boldsymbol{h}$: A tensor of vectors that representing embeddings. |
| | $\nabla_{out}$ : The derivatives of the loss function with respect to $\int_0^x f(t; \boldsymbol{h})\, \mathrm{d}t$ for all $x$. |
| *Outputs:* | $\nabla_x$: The gradient of $\int_0^x f(t; \boldsymbol{h})\, \mathrm{d}t$ with respect to x. |
| | $\nabla_{\boldsymbol{\theta}}$: The gradient of $\int_0^x f(t; \boldsymbol{h})\, \mathrm{d}t$ with respect to f parameters. |
| | $\nabla_{\boldsymbol{h}}$: The gradient of $\int_0^x f(t; \boldsymbol{h})\, \mathrm{d}t$ with respect to h. |
| *Hyper-parameters:* | $f$: A derivable function $\mathbb{R} \to \mathbb{R}$. |
| | $N$: The number of integration steps. |

1: **procedure** BACKWARD($x$, $h$, $\nabla_{out}$; $f$, $N$)
2:     ▷ Compute weights and evaluation steps for Clenshaw-Curtis quadrature
3:     $\boldsymbol{w}, \boldsymbol{\delta}_x = $ COMPUTE_CC_WEIGHTS($N$)
4:     $F, \nabla_{\boldsymbol{\theta}}, \nabla_{\boldsymbol{h}} = 0, 0, 0$
5:     **for** $i \in [1, N]$ **do**
6:         $x_i = x_0 + \frac{1}{2}(x - x_0)(\boldsymbol{\delta}_x[i] + 1)$                    ▷ Compute the next point to evaluate
7:         $\delta_F = f(x_i; \boldsymbol{h})$
8:         ▷ Sum up for all samples of the batch the gradients with respect to inputs $\boldsymbol{h}$
9:         $\delta_{\nabla_{\boldsymbol{h}}} = \sum_{j=1}^B \nabla_{\boldsymbol{h}^j} \left(\delta_F^j\right) \nabla_{out}^j (x^j - x_0^j)$
10:         ▷ Sum up for all samples of the batch the gradients with respect to parameters $\boldsymbol{\theta}$
11:         $\delta_{\nabla_{\boldsymbol{\theta}}} = \sum_{j=1}^B \nabla_{\boldsymbol{\theta}} \left(\delta_F^j\right) \nabla_{out}^j (x^j - x_0^j)$
12:         $\nabla_{\boldsymbol{h}} = \nabla_{\boldsymbol{h}} + \boldsymbol{w}[i]\delta_{\nabla_{\boldsymbol{h}}}$
13:         $\nabla_{\boldsymbol{\theta}} = \nabla_{\boldsymbol{\theta}} + \boldsymbol{w}[i]\delta_{\nabla_{\boldsymbol{\theta}}}$
14:     **end for**
15:     ▷ Gradients with respect to superior integration bound.
16:     $\nabla_x = f(x, \boldsymbol{h})\nabla_{out}$
17:     **return** $\nabla_x, \nabla_{\boldsymbol{\theta}}, \nabla_{\boldsymbol{h}}$
18: **end procedure**

---

## C   Generated images from MNIST

Figure 4 presents samples generated from two UMNN-MAF trained on MNIST, respectively with (sub-figure a) and without (sub-figure b) labels. The samples are generated with different levels of noise, which are the product of the inversion of the network with random values drawn from $\mathcal{N}(0, T)$, with $T$ being the sampling temperature. The sampling temperature increases linearly from $0.1$ (top rows) to $1.0$ (bottom rows). We can observe that the unconditional model fails to incorporate digit structure when the level of noise is too small. However, when the level is sufficient it is able to generate random digits with a high level of heterogeneity.

(a)                                    (b)

Figure 4: (a): Class-conditional generated images from MNIST. The temperature of sampling increases from $0.1$ (top row) to $1.0$ (bottom row). Columns correspond to different classes. (b): Unconditional generated images from MNIST. The temperature of sampling goes from $0.1$ at top row to $1.0$ at bottom row. Columns are different random noise values.