[Reviews · NeurIPS 2019]

Reviewer 1



*** Update in response to author rebuttal *** I definitely think that the UMNN technique has good potential and the authors should continue to pursue it. However, even after reading the rebuttal, I feel that it is a bit premature to publish the research at this point in time. In the rebuttal, the authors acknowledge that their method is not the first universal monotonic approximator and clarify that their language regarding the "cap on expressiveness" of alternative monotonic approximators refers to the non-asymptotic case, i.e., a finite number of neurons/hidden units. They write "we believe that the constraints on the positiveness of the weights and on the class of possible activation functions are unnecessarily restraining the hypothesis space in the non-asymptotic case". However, this is an assertion for which they have not supplied any kind of proof, and I find it highly debatable. Any method, whether it is their UMNN or the Huang approach or lattices or max/min networks, has some cap on expressiveness in the non-asymptotic case. They seem to be suggesting that given a fixed budget of parameters (neurons or whatever), that the UMNN technique will be more expressive (i.e., model monotone transformations more accurately) than alternative techniques, but it's not obvious at all to me that this is true. I am particularly skeptical given that UMNN (unlike other techniques) requires numerical integration which itself necessarily involves some sort of imperfect approximation given a finite number of integration steps. Even setting aside numerical integration, it is not clear to me that a UMNN with a a budget of P parameters will be more expressive than a lattice or max/min network allocated the same number of parameters P. Lines 148-149 in the paper are a better description of the advantages of the technique: "it enables the use of any state-of-the-art neural architecture". This is where the potentially significant advantage of the UMNN method is, in my opinion. Sometimes certain architectures or activation functions work better on certain problems than others for somewhat mysterious reasons (local minima using one activation function which do not occur using another one, etc) and their technique allows for the possibility of trying various different activation functions (rectified linear, sigmoidal, RBF, etc) and other architecture variations. Unfortunately, they don't illustrate this advantage experimentally. If they had shown that sigmoidal UMNN fails on one dataset whereas ReLU does well but on a second dataset ReLU fails but sigmoidal does well, that would be a nice illustration of the flexibility advantages of their technique, but these kinds of experiments are not in the paper. I suggest the authors pursue this line of research in the future. As it is, what they have presented seems to be mostly just a new way to do monotonic modeling but one that is not obviously better- the "cap on expressiveness" language is entirely unconvincing to me. Also, their claim that NAF and B-NAF did not report results on MNIST due to memory explosion still seems to me to be speculation. It shouldn't be that hard to calculate how many parameters (and therefore how much memory) NAF or B-NAF would actually require on MNIST- you could almost surely do the memory requirement calculations without actually implementing the techniques. As I look at the number of parameters table (Table 2), I see 7.49e6 parameters for NAF for MINIBOONE (d=43), 3.46e6 for UMNN, which is a ratio of 749/346=2.16. For BSDS300, it is 3.68e7 vs. 1.56e7, a ratio of 2.36, not much bigger than 2.16 for MINIBOONE. It's not clear to me how to extrapolate the ratio for MNIST, but it's not at all obvious that the ratio would be that large- it might only be a factor of 3 or 4, which is disadvantageous but hardly explosion. If the ratio is much higher than 3 or 4, well, the authors had an opportunity to tell me the actual ratio after I nudged them, but they declined to do so. Again, I also don't think the experimental results are that strong. For VAE, it loses across the board to B-NAF. For density estimation, it looks like it also loses by a wider margin on GAS and MINIBOONE than the margin by which it wins on POWER, and it seems questionable to claim that it wins on MNIST when the other techniques simply do not report on MNIST. *** Original comments *** I have good familiarity with the monotonic modelling literature and I have not seen any prior work which models the derivative and then integrates, so this technique definitely looks original to me. One significant concern I have about the paper is that it does not do a good job of describing prior work on monotonic modeling. The authors claim that (prior to their own work) architectures which enforce monotonicity do so in ways which "lead to a cap on the expressiveness" of the hypothesis class. This is not correct regarding the Sill, 1998 "Monotonic Networks" paper. That paper shows that a 2-hidden-layer architecture which combines minimum and maximum operations is capable of universal approximation of monotonic functions, given sufficient hidden units. The authors also fail to cite the work of Gupta et. al. on lattice-based monotonic models ("Monotonic Calibrated Interpolated Lookup Tables", JMLR 2016 and "Deep Lattice Networks and Partial Monotonic Functions", NeurIPS 2017). The authors claim "[o]ur primary contribution consists in a neural network architecture that enables learning arbitrary monotonic functions". It is true that they present a novel way to do so, but there are multiple established techniques already in the literature for this. So while the work is original, the significance is overstated. A stronger paper would have compared their "implement the derivative and integrate" technique to the other techniques or at least attempted to explain why their technique is preferable. Another concern is that the experimental results are respectable but not as strong as I would like. UMNN-NAF only wins on 2 of the 6 density estimation tasks. In variational auto-encoding, it loses to FFJORD and B-NAF. It is indeed enocuraging that the authors obtained results on MNIST, and it might indeed be the case that competing techniques cannot run on MNIST due to "memory explosion", but this appears to be a speculative theory on the part of the authors. The authors would be on firmer ground if they had actually attempted to implement competing techniques on MNIST and found memory explosion, but they do not appear to have done so.

Reviewer 2



The authors well formulated the task associated to the unconstrained monotonic neural networks, and accordingly demonstrated the technical outline to design a neural network architecture that enables learning arbitrary monotonic functions. There the forward integration and backward integration are clearly explained. In addition, the authors demonstrated some UMNN autoregressive models, typically and yet interestingly including those related to the normalizing flows and autoregressive transformations, which may help us render new methods to solve many traditional hard tasks in information processing, image processing, and machine learning. In a summary, its novelty and importance are both significant, in my option.

Reviewer 3



# Originality The authors propose to parameterize a monotonic network by parameterizing its derivative. The derivative of a monotonic network is much easier to parameterize because it's only required to be positive, and thus can be modeled by "free-form" neural networks, without special constraints on the weights and activation functions. The monotonic network can then be computed by numerical integration of the derivative network. Backpropagation through the monotonic network can be done by another numerical integration. For all numerical integration computations, this paper proposes to use the Clenshaw-Curtis quadrature approach, which is efficiently parallelizable. The monotonic networks proposed are used for constructing an expressive autoregressive flow model, which is proved to be competitive to NAFs and B-NAFs. The approach is novel. The difference from previous work is clear. Closely related works such as NAF and B-NAFs are cited and compared. I also like the explicit mentioning of numerical inversion of autoregressive flow models and the experiments to demonstrate the samples. Although the approach is straightforward, this is probably the first time that samples have been reported for these autoregressive flow models. # Quality The submission is technically sound. There are also some discussions on the limitations of this method in section 6. The experimental evaluation part is also largely satisfying, and it would benefit from clarifying the following: 1. In section 5.2, the authors hypothesized that NAF and B-NAF do not report results on MNIST due to memory explosion. I'm wondering why memory should be an issue for NAFs and B-NAFs to work well on MNIST? Why can using UMNN solve this problem? There doesn't seem to be any inherent difference of UMNN that makes it particularly good for memory limited cases. 2. Numerical integration is arguably slow and only provides an approximation, yet it is used for both forward and backward propagation of the network. How fast is the Clenshaw-Curtis quadrature algorithm? I noticed that the authors didn't check the "An analysis of the complexity" item on the reproducibility checklist. It would be better to explicitly discuss this somewhere in section 2 or 3. Also, accurate integration requires the derivative networks to have small Lipschitz constants. Do you have the results on performance vs different Lipschitz constant and different number of integration steps? Both NAF and B-NAF showed some theoretical results on uniform density estimators. Is UMNN also a uniform density estimator? I would imagine so, but it would be nice to have some formal proof and discussion on this. # Significance As the results are reasonable and the approach is novel, I think this work provides an interesting and useful idea for the field. # Clarity The paper is very well written.

[Author Response · NeurIPS 2019]

We would like to thank the reviewers for their careful reading and positive assessment of our work (**Rev #1**: "*this technique definitely looks original*", **Rev #2**: "*novelty and importance are both significant*", **Rev #3**: "*this work provides an interesting and useful idea to the field*"). We attempt hereafter to address the main concerns raised by the reviewers.

**Related works (Rev #1).** **Rev #1** points out the lack of references to You et al. [2017] and Gupta et al. [2016] which combine deep neural networks (NN) with monotonic lattice regression in order to learn functions that are monotonic with respect to a subset of their input variables. We thank the reviewer for these references, they are very relevant and will be added to the manuscript. **Rev #1** also suggests to add experiments in order to compare UMNNs with this method. In our opinion, a complete review of monotonic NNs and their use in the context of normalizing flows (NF) are very relevant and certainly worth of many valuable insights, but should be carried out within the scope of an extended or separate paper. Proposing a new parametric monotonic transformation and exploring how to combine it with autoregressive architectures into a NF is already a significant contribution in itself (**Rev #1**: "*This paper contributes a novel technique for modeling monotonic functions [...] that is a significant contribution*" **Rev #1**: "*The technique is applied to autoregressive flows [...] shown to have competitive performance results*" **Rev #3**: "*A new way of parameterizing monotonic networks [...] this is significant, and can inspire more future work*"). Given the page limit constraints, we are afraid that adding experiments in the current manuscript would decrease its clarity and concision.

**Universality of UMNNs (Rev #1, #3).** We thank **Rev #1** to have pointed out the ambiguity of our statement about the difference in terms of expressiveness between UMNNs and previous monotonic neural networks. We do not want to erroneously claim that UMNNs are the first universal approximator of monotonic transformations. Instead, we argue that other neural architectures for density estimation do so in a way that "leads to a cap on the expressiveness" of the transformations in the non-asymptotic case (finite number of neurons). While Sill [1998] (as well as Huang et al. [2018] and De Cao et al. [2019] for universal *density* approximators) has proven the universality of his approach in the asymptotic case, we believe that the constraints on the positiveness of the weights and on the class of possible activation functions are unnecessarily restraining the hypothesis space in the non-asymptotic case. We will make sure to clarify our statement in the next version of the manuscript. On a similar track, **Rev #3** wonders if UMNN is a uniform density estimator. Yes, for continuous random variables. By relying on the inverse sampling theorem it is enough to prove that UMNNs are universal approximators of continuously derivable ($C^1$) monotonic functions. Indeed, if UMNNs can represent any $C^1$ monotonic function, then they can also represent the (inverse) cumulative distribution function of any continuous random variable. Any continuously derivable function $f : \mathcal{D} \to \mathcal{I}$ can be expressed as the following integral: $f(x) = \int_a^x \frac{df}{dx} dx + f(a), \quad \forall x, a \in \mathcal{D}$. The derivative $\frac{df}{dx}$ is a continuous positive function and it is known that this function can be successfully approximated by a NN (such as those used in UMNNs) thanks to the universal approximation theorem of NNs.

**Theory: Scalability and complexity analysis (Rev #1, #2, #3).** **Rev #1** and **#3** show concerns regarding the superior scalability (in terms of memory) of UMNNs in comparison to NAF and B-NAF. UMNNs are particularly well suited for density estimation because the computation of the Jacobian only requires a single forward evaluation of a NN. Together with the Leibniz integral rule, they make the evaluation of the log-likelihood derivative as memory efficient as usual supervised learning, which is equivalent to a single backward pass on the computation graph. By contrast, density estimation with previous monotonic transformations typically requires a backward evaluation of the computation graph of the transformer NN to obtain the Jacobian. Then, this pass must be evaluated backward again in order to obtain the log-likelihood derivative. Both NAF and B-NAF provide a method to make this computation numerically stable, however both fail at not increasing the size of the computation graph of the log-likelihood derivative, hence leading to a memory overhead. **Rev #3** also asked about the speed of Clenshaw-Curtis algorithm. In the case of static Clenshaw-Curtis, the function values at each evaluation point can be computed in parallel using batch of points. Thus, the limitation comes usually from the GPU memory which might not be large enough to store "meta-batches" of size $d \times N \times B$ (with $d$ the dimension of the data, $N$ the number of integration steps and $B$ the batch size). **Rev #3** also asked about the relation between Lipschitzness and number of integration steps. We did not formally assess the impact of the number of integration steps and Lipschitz constant. However, we observed that as long as the Lipschitz constant of the network does not explode ($< 1000$), a reasonable number of integration steps ($< 100$) is sufficient to ensure the convergence of the quadrature. **Rev #2** suggests to add a discussion about the design of the NNs. We would like to recall that we provide all the experimental details in the appendix, moreover the code will be publicly released. Finally, **Rev #2** would be in favor of a deeper theoretical discussion. **We will make sure to develop and clarify all these minor elements in the revised version of the manuscript.** We will take advantage of the discussion about Lipschitzness to provide some insights about the design of the different neural networks.

**More experiments (Rev #2).** More density estimation experiments are suggested by **Rev #2**. We agree with him/her that more experiments are always a plus and can only improve the quality of our work. However we would like to bend the fact that we used the classical benchmarks for NFs (**Rev #3**: "*The experimental evaluation part is also largely satisfying*"). We even did more experiments than most of the competing methods (**Rev #3**: "*this is one of the earliest works (if not the first) that directly inverts an autoregressive flow*").

[Meta-Review · NeurIPS 2019]

The paper introduces a new approach to model monotonic functions by integrating the output of a non-negative neural network. All reviewers thought the idea is interesting, original, and worth pursuing. In particular, it does seem to offer more flexibility compared to existing approaches. Please revise some of the claims on the expressiveness (compared to existing work) as suggested by the reviewers. If possible, more thorough experiments could better demonstrate the effectiveness of the approach (reviewer 1 provided some great suggestions).